# Ethnic-Specific and UV-Independent Mutational Signatures of Basal Cell Carcinoma in Koreans

**DOI:** 10.3390/ijms26146941

**Published:** 2025-07-19

**Authors:** Ye-Ah Kim, Seokho Myung, Yueun Choi, Junghyun Kim, Yoonsung Lee, Kiwon Lee, Bark-Lynn Lew, Man S. Kim, Soon-Hyo Kwon

**Affiliations:** 1Translational-Transdisciplinary Research Center, Clinical Research Institute, Kyung Hee University Hospital at Gangdong, Kyung Hee University College of Medicine, Seoul 05278, Republic of Korea; yeak426@khu.ac.kr (Y.-A.K.); tjzh13@khu.ac.kr (S.M.); uag43@khu.ac.kr (Y.C.); ylee3699@khu.ac.kr (Y.L.); 2Department of Biomedical Science and Technology, Graduate School, Kyung Hee University, Seoul 02447, Republic of Korea; 3Department of Medicine, Kyung Hee University College of Medicine, Seoul 02453, Republic of Korea; 4Division of Tourism & Wellness, Hankuk University of Foreign Studies, Yongin-si 17035, Republic of Korea; jh.kim@hufs.ac.kr; 5Department of Bioscience and Biotechnology, Hankuk University of Foreign Studies, Yongin-si 17035, Republic of Korea; leekw@hufs.ac.kr; 6Department of Dermatology, Kyung Hee University Hospital at Gangdong, Kyung Hee University College of Medicine, Seoul 02453, Republic of Korea; bellotte@hanmail.net

**Keywords:** basal cell carcinoma, whole-genome sequencing, skin cancer

## Abstract

Basal cell carcinoma (BCC), the most common skin cancer, is primarily driven by Hedgehog (Hh) and TP53 pathway alterations. Although additional pathways were implicated, the mutational landscape in Asian populations, particularly Koreans, remains underexplored. We performed whole-exome sequencing of BCC tumor tissues from Korean patients and analyzed mutations in 11 established BCC driver genes (*PTCH1*, *SMO*, *GLI1*, *TP53*, *CSMD1/2*, *NOTCH1/2*, *ITIH2*, *DPP10*, and *STEAP4*). Mutational profiles were compared with Caucasian cohort profiles to identify ethnicity-specific variants. Ultraviolet (UV)-exposed and non-UV-exposed tumor sites were compared; genes unique to non-UV-exposed tumors were further analyzed with protein–protein interaction analysis. BCCs in Koreans exhibited distinct features, including fewer truncating and more intronic variants compared to Caucasians. Korean-specific mutations in *SMO*, *PTCH1*, *TP53*, and *NOTCH2* overlapped with oncogenic gain-of-function/loss-of-function (GOF/LOF) variants annotated in OncoKB, with some occurring at hotspot sites. BCCs in non-exposed areas showed recurrent mutations in *CSMD1*, *PTCH1*, and *NOTCH1*, suggesting a UV-independent mechanism. Novel mutations in *TAS1R2* and *ADCY10* were exclusive to non-exposed BCCs, with protein–protein interaction analysis linking them to *TP53* and *NOTCH2*. We found unique ethnic-specific and UV-independent mutational profiles of BCCs in Koreans. *TAS1R2* and *ADCY10* may contribute to tumorigenesis of BCC in non-exposed areas, supporting the need for population-specific precision oncology.

## 1. Introduction

Basal cell carcinoma (BCC) is the most common skin cancer subtype, and its incidence is rapidly increasing worldwide [1]. Although BCC rarely metastasizes or affects mortality, it can cause substantial morbidity through local invasion and tissue destruction. Asians with Fitzpatrick skin phototypes (FSPT) III to IV are at lower risk of BCC than Caucasians with FSPT phototypes I to II. However, BCC incidence in Korea was 7.6 per 100,000 person/year in 2019, representing 7.6-fold increase over the last 20 years [2]. Due to an aging population, increased outdoor activity, and continuing depletion of the ozone layer, the rapid rise in BCC incidence is expected to worsen over the next few decades.

Aberrant activation of the sonic hedgehog (Hh) signaling pathway plays a crucial role in BCC [3]. In physiological skin, the Hh pathway is implicated in maintaining the stem cell population and regulating hair follicle and sebaceous gland development. However, mutations that inactivate patched 1 (*PTCH1*) or activate smoothened (*SMO*) result in the translocation of activated glioma-associated transcription factor 1 (*GLI1*) into the nucleus. The expression of *GLI1*-targeted genes, such as *CCND1/CCND2*, *MYC*, *TWIST*, Wnt pathway, *BCL2*, and *FOXM1*, affects cell proliferation, cell cycle regulation, apoptosis, angiogenesis, and self-renewal, causing BCC [3]. *PTCH1* inactivation is the most frequently mutation in BCC, reported in 11–75% of sporadic BCCs [4,5,6,7,8,9,10], while activating mutations of SMO occur in 10–20% of BCCs [9,10,11]. Meanwhile, inactivating mutations of the *TP53* tumor suppressor gene have been reported in approximately half of sporadic BCCs [5,7,9,10,12,13,14,15,16,17]. In addition to regulating cell cycle arrest and programmed cell death, the inactivation of *TP53* activates the Hh pathway by upregulating *SMO* [18].

Chronic exposure to ultraviolet (UV) radiation is the primary environmental risk factor and a major driver of mutagenesis in BCC. UV radiation induces C-to-T or CC-to-TT mutations via cyclobutane dimers and pyrimidine (6-4) pyrimidine photoproducts [19,20]. When these signature mutations occur in oncogenes or tumor suppressor genes, BCCs may develop. BCC typically appears on the sun-exposed skin of older adults, particularly on the face and neck. However, a significant proportion (17–26% in Caucasians [21,22], 7–8% in Koreans [23]) of BCCs occur in non-exposed areas such as the trunk or genitalia. While the nodular subtype predominates in the head and neck, superficial BCCs occur predominantly in non-exposed areas [21]. Patients with non-exposed BBCs were younger and had larger tumors than those with head and neck BCCs [21].

Although the activation of the Hh signaling pathway is a defining feature of BCC, recent genomic studies have revealed that additional driver mutations in cancer-related genes are involved. A large-scale study reported that 85% of BCCs harbored at least one mutation other than *PTCH1*, *TP53*, and *SMO* [24]. Interestingly, tumor location and UV exposure influenced these mutational patterns. In a previous study of Caucasian patients, *PTCH1* mutations were significantly associated with sun-exposed areas, whereas *NOTCH1* mutations were more frequent in non-sun-exposed regions [24]. These findings suggested that UV-related mutagenesis contributed to regional heterogeneity in BCC pathogenesis.

In addition to anatomical differences, ethnic variations in the genomic landscape of BCC have been proposed [25]. However, large-scale genomic studies have focused predominantly on Caucasian populations, and the mutational profile of BCCs in Asian patients remains underexplored.

In the present study, we aimed to investigate the mutational landscape of BCCs in a Korean cohort and compare these findings with published data from Caucasian patients. We analyzed a panel of 13 BCC marker genes previously identified in an Italian study [24], including *PTCH1*, *SMO*, *GLI1*, *CSMD1*, *CSMD2*, *NOTCH1*, *NOTCH2*, *TP53*, *ITIH2*, *DPP10*, and *STEAP4*, as well as promoter mutations in *TERT* and *DPH3*. However, due to inconsistent annotations and ambiguous genomic coordinates, *TERT* and *DPH3* promoter mutations were excluded from our analysis. Furthermore, we aimed to identify distinct driver mutations in non-sun-exposed tumors compared with sun-exposed tumors, particularly focusing on Korean patients.

## 2. Results

### 2.1. Study Cohort

Fourteen sporadic BCCs from fourteen patients (nine males and five females) were included in the study. The mean age at diagnosis was 67.50 ± 8.06 (range, 51–79 years). Demographic features of the patients and clinicopathologic characteristics of the BCCs are described in Table 1.

### 2.2. Mutations in Caucasians and Koreans Reveal Different Patterns in BCC

To elucidate the genetic alterations underlying BCC in a Korean population, we performed whole-exome sequencing of 11 previously implicated genes and compared the resulting mutation profiles with those reported in the Caucasian cohort [24]. The analyzed genes included Hh pathway genes *PTCH1*, *SMO*, *GLI1*, *CSMD1*, *CSMD2*, *NOTCH1*, *NOTCH2*, *TP53*, *ITIH2*, *DPP10*, and *STEAP4*.

*PTCH1*, a tumor suppressor gene associated with the Hh pathway, harbored Korea-specific mutations, including two missense mutations, four splice mutations, three truncating mutations, and one intron mutation, with mutations located in sterol-sensing domain and a patched family functional protein domain (Figure 1A). Compared with the Caucasian dataset [24], the Korean cohort lacked several *PTCH1* alterations, including one in-frame mutation, nineteen splice mutations, seventeen missense mutations, thirteen intron mutations, and thirty-nine truncating mutations.

*SMO*, an oncogene in the Hh pathway, displayed a single missense mutation within the frizzled family functional domain that was unique to the Korean cohort (Figure 1B). In contrast, the Caucasian dataset harbored additional *SMO* mutations, two splice mutations, four missense mutations, two truncating mutations, and nine intron mutations that were not identified in Koreans. Meanwhile, *GLI*, another known oncogene in the Hh pathway, exhibited no novel coding mutations and only two intron mutations specific to the Korean cohort. Meanwhile, seven missense, seventeen truncating, one splice, and four intron mutations were missing compared to Caucasian *GLI* coding variants (Figure 1C).

*TP53* showed two missense mutations and one splice mutation within its DNA-binding domain in the Korean cohort (Figure 1D). However, the Korean cohort lacked the eight missense mutations, ten truncating mutations, and five splice mutations, or thirteen intron mutations identified in the Caucasian cohort. Most other analyzed genes (*CSMD1*, *CSMD2*, *NOTCH1*, *NOTCH2*, *ITIH2*, and *DPP10*) followed a similar pattern; a few novel alterations were specific to the Korean cohort, whereas numerous Caucasian variants were absent (Appendix A). Only *STEAP4* showed no mutations in the Korean cohort.

### 2.3. BCCs in Non-Exposure Areas Exhibited Significant Mutations in CSMD1, PTCH1 and NOTCH1

We conducted additional analysis to compare the genetic mutations based on the anatomical sites: UV-exposed and non-exposed areas. We identified different mutation frequencies in targeting BBC marker genes between these groups. SNP analysis revealed that the UV-exposed group harbored more mutations in these genes than the non-exposed group. *CSMD1*, *CSMD2*, *PTCH1*, *GLI1*, and *NOTCH1* showed different mutation positions in both the exposed and non-exposed groups, whereas the others showed different mutated SNPs in the exposed group (Figure 2). These five genes exhibited higher mutation prevalence in the exposed groups.

*CSMD1* displayed one missense mutation in the non-exposed group, while the exposed group showed missense and truncating mutations in the exon. However, the mutation in the non-exposed *CSMD2* group was located in the intron, which was less important. *PTCH1* showed a truncating mutation in the sterol-sensing domain, and another mutation was located in an exon associated with a patched function in the exposed group. Exploring mutations within *SMO*, *GLI1*, and *ITIH2* revealed a solitary missense mutation in *SMO* of the exposed group, while no mutations were detected in the exons of *GLI1* in either group or *ITIH2* in the exposed group. *NOTCH1* included one missense mutation in the non-exposed group, and three missense mutations in the exposed group. However, *NOTCH2*, *TP53*, and *DPP10* displayed mutations only in the exposed group, with *TP53* exhibiting missense variants in the *P53* domain, which is involved in tumor suppression [26] (Appendix A). Therefore, *CSMD1*, *PTCH1*, and *NOTCH1* manifested amino acid-altering mutations in the exons of the non-exposed group.

### 2.4. Mutations of ADCY10 and TAS1R2 Found in the Exposed Areas Interact with Ith BCC Marker Genes

To further explore the difference between the BCCs in exposed and non-exposed areas regarding the significantly mutated genes, we conducted SNP analysis independent of the 11 previously studied BCC marker genes. We identified missense variant mutations in *ADCY10* and *TAS1R2* in the non-UV-exposed group compared to the UV-exposed group. The frequencies of SNP alleles in the non-UV-exposed group for *TAS1R2* and *ADCY10* were 0.4167 and 0.3, respectively, while these variants were absent in controls (Appendix A). Given the limited prior research on these genes, protein–protein interaction (PPI) analysis was conducted between BCC marker genes and the selected genes. The PPI network, filtered with a confidence of 0.15, is presented in Figure 3, which shows the connections between *ADCY10*, *TAS1R2* and BCC marker genes. Enriched functions from the network were identified by STRING [27] using Gene Ontology biological processes and included functions associated with epithelial cell fate and growth—epithelial development, epithelial cell proliferation, epithelial cell differentiation, epidermal cell fate specification, and cellular process—as well as positive regulation of apoptosis, positive regulation of cellular process, and regionalization. Among BCC marker genes, *PTCH1*, *GLI1*, *TP53*, *SMO*, *NOTCH1*, and *NOTCH2* interacted with each other and shared common enriched functions, including epithelial development, positive regulation of cellular processes, and regionalization. *ADCY10* and *TAS1R2* are involved in the positive regulation of cellular processes, whereas *ADCY10* is also related to the positive regulation of apoptosis by *TP53*. Interestingly, *TP53* and *NOTCH2* displayed experimentally determined connections with *ADCY10* and *TAS1R2*.

## 3. Discussion

Recent genomic studies identified novel genetic alterations beyond the Hh pathway, contributing to pathogenesis of BCC. *CSMD1* and *CSMD2*, which encode inhibitors of the complement system, have been proposed as tumor suppressor genes, with *CSMD1* being the second most frequently mutated gene in BCC [24,28,29]. LOF mutations in *NOTCH1*, observed in approximately 30–50% of BCCs, are considered to facilitate tumor persistence in sporadic cases [10,30]. Non-coding mutations in the TERT promoter, present in 39–74% of BCCs, are associated with telomere shortening and subsequent telomerase activation, a well-established hallmark of numerous human cancers [31,32,33,34]. Other genes that are frequently mutated in BCC include *LATS1* and *PTPN14*, which are critical components of the Hippo-YAP signaling pathway [10], the *DPH3* promoter [34], *MYCN* [10], *PPP6C* [10], and *STK19* [10]. These findings suggest that BCC harbors a heterogeneous genetic landscape involving a broader network of cancer-associated genes than was previously recognized.

In our analysis, we focused on the 11 genes, including Hh pathway genes and additional genes with mutations previously reported in the Caucasian BCC cohort [24]. We specifically sought to identify mutations that were unique to the Korean cohort. We observed the L412F mutation in SMO, a known cancer hotspot that likely confers a GOF oncogenic effect via aberrant Hh pathway activation. Within *PTCH1*, several variants, 1183, V1083Wfs4, X844_splice, L490, R459Lfs49, X195_splice, and X132_splice, were predicted to be oncogenic LOF mutations according to OncoKB, further supporting the central role of Hh signaling in BCC pathogenesis in the Korean population. We detected three cancer hotspot mutations in *TP53*, V274D, R175H, and X126_splice, which potentially resulted in LOF due to OncogenicDB. These findings highlight the critical role of *TP53* in the Korean BCC population compared to its established significance in other malignancies [35,36].

Additionally, a potentially oncogenic LOF mutation (OncogenicDB) in *NOTCH2* (Q1061) was identified, suggesting that signaling pathways beyond the Hh and *TP53* axes may contribute to BCC pathogenesis. *NOTCH2* plays a crucial role in epidermal cell differentiation, and its dysregulation has been implicated in BCC, highlighting its role in skin homeostasis and tumorigenesis [37]. While these pathogenic variants correspond to established oncogenic drivers, several Korean-specific mutations remain of uncertain clinical significance (not annotated in OncogenicDB). Further functional and epidemiological studies are needed to elucidate their tumorigenic potential and refine our understanding of population-specific genetic variations in BCC (Appendix A).

To further investigate population-specific genetic differences, we analyzed the proportion of mutations unique to Korean and Caucasian cohorts. Intronic mutations represented the most abundant variant class in both cohorts, comprising 54.7% and 31.9% of Korean-specific and European-specific mutations, respectively. The significantly elevated proportion of intronic variants in the Korean cohort suggests potential population-specific differences in gene regulation or splicing efficiency, as intronic mutations may influence gene expression and protein function despite being traditionally classified as noncoding [38]. Missense mutations were less prevalent in the Korean-specific mutation group (17.8%) than in the European-specific group (21.4%). Similarly, mutations in the splice region were observed at a slightly higher frequency in the Korean-specific mutation group (10.6%) than in the European-specific mutation group (9.8%).

A notable difference was observed in the truncating mutations. In the Korean-specific mutation group, only 6.84% of the mutations were truncating, and the majority were nonsense mutations. In contrast, truncating mutations were significantly more prevalent in the Caucasian dataset (26.3%), dominated by frameshift insertions (FS ins) and deletions (FS del). These findings highlight population-specific differences in BCC in the mutational spectrum, where the Caucasian cohort exhibited a higher prevalence of FS ins and FS del, whereas the Korean cohort was more concentrated in intronic regions. Further studies are needed to assess whether these genomic disparities influence disease progression, treatment responses, and clinical presentations.

UV exposure is a well-established etiological factor in the development of skin cancers, particularly BCC. Genetic alterations resulting from UV-induced DNA damage play a critical role in BCC pathogenesis. Additionally, while the impact of UV exposure and BCC occurrence regarding systemic implications was presented by previous research, the mechanism beyond BCC in non-UV-exposed areas is not well-known [39]. In the previous study of Caucasian patients, *PTCH1* mutations were significantly associated with sun-exposed areas, whereas *NOTCH1* mutations were more frequent in truncal BCCs, a typical non-exposed region [24]. These findings demonstrate that BCC from different anatomical sites may exhibit distinct clinical, histological, and molecular profiles, highlighting BCC heterogeneity and the need to further characterize mutations in non-exposed tumors.

In our comparative analysis of UV-exposed and UV-unexposed BCC samples, we found that BCCs in exposed areas harbored a greater number of SNPs and pathogenic mutations. However, mutations in *CSMD1*, *CSMD2*, *PTCH1*, *GLI1*, and *NOTCH1*—all recognized BCC marker genes—were identified in non-UV-exposed groups. *CSMD1*, *PTCH1*, and *NOTCH1* exhibited missense or truncating mutations that may alter protein structure or function. This suggested that *CSMD1* and *PTCH1* play significant roles in BCC pathogenesis of BCCs in non-exposed areas. Moreover, *PTCH1* mutations in the non-exposed area included splice-site (X844_splice) and nonsense mutation (L490*, Q365*, and Q84*) variants (Appendix A). While *PTCH1* mutations are well-known in UV-induced carcinogenesis [40,41], our results imply they may also contribute to BCC carcinogenesis independent of UV exposure. *PTCH1* and *NOTCH1* are key components of the Hh and *NOTCH* signaling pathways, respectively, highlighting their critical roles in BCC. Additionally, *PTCH1* contributed to BCC development in a Caucasian cohort study [24].

In the PPI network between *TAS1R2*, *ADCY10* and BCC marker genes, the core six genes—*NOTCH1*, *NOTCH2*, *TP53*, *GLI1*, *PTCH1*, and *SMO*—were interconnected. Specific interactions were observed between *NOTCH2* and *TAS1R2*, and *TP53* and *ADCY10*. All mutated genes from the non-UV-exposed groups were included in the PPI network, exhibiting edges between *CSMD1* and *NOTCH1*, which share functional enrichment in epithelial development. While *TAS1R2* has been associated with solid tumors in previous studies [42,43], its role in BCC progression and carcinogenesis remains unclear. Its interaction with *NOTCH2* indicates its potential role in BCC pathogenesis. *ADCY10* mutations have been identified in multiple cancers, including Merkel cell carcinoma and breast cancer [44,45]. Interestingly, a previous study has found that adenylyl cyclase, particularly *ADCY10*, is linked to ferroptosis in lung cancer [46]. Additionally, soluble adenylyl cyclase has been implicated in hyperproliferative skin disease, even though it was similar to normal human skin in BCC [47].

Our study has several limitations. First, while we identified mutations unique to the Korean cohort, functional validation was not performed to confirm their oncogenic effects. Second, our study relied on SNP-based analysis, which may not fully capture structural variants or epigenetic modifications involved in BCC progression. Lastly, the sample size was limited, requiring further large-scale studies to confirm our findings.

## 4. Materials and Methods

### 4.1. Sample Collection

BCC tumor tissues (diameter, 5 mm) and matched peripheral blood samples were obtained from the fourteen patients. This study was conducted in accordance with the Declaration of Helsinki and the International Conference on Harmonization and Good Clinical Practice Guidelines and was reviewed and approved by the Institutional Review Board of Kyung Hee University Hospital at Gangdong (KHNMC 2022-04-014). Written informed consent was obtained from all participants before enrolment in the study.

### 4.2. DNA Extraction from Tumor Samples

We utilized the Agilent SureSelect Target Enrichment protocol for Illumina Paired-End Sequencing Library (Version C2, December 2018) with 0.5 µg input gDNA to generate standard exome capture libraries. All libraries were enriched using SureSelect Human All Exon V6 Probe (Agilent, Santa Clara, CA, USA). DNA quantification and quality were assessed using PicoGreen (Invitrogen, Waltham, MA, USA) and Tapestation gDNA Screentape (Agilent, Santa Clara, CA, USA). We used 1 μg genomic DNA diluted in EB Buffer and sheared to a target peak size of 150–200 bp using the Covaris LE220 focused-ultrasonicator (Covaris, Woburn, MA, USA) according to the manufacturer’s protocol. The DNA was sheared in an 8-microTUBE Strip (Covaris, Woburn, MA, USA) into the tube holder of the ultrasonicator using the following settings: mode, frequency sweeping; duty cycle, 10%; intensity, 5; cycles per burst, 200; duration, 60 s × 6 cycles; temperature, 4–7 °C.

The fragmented DNA was repaired and underwent A-tailing to the 3′ end, followed by ligation of Agilent adapters to the fragments. After verifying ligation, the adapter-ligated product was amplified using PCR. For exome capture, 250 ng of the DNA library was combined with hybridization buffers (Agilent, Santa Clara, CA, USA), blocking mixes (Agilent, Santa Clara, CA, USA), RNase block (Agilent, Santa Clara, CA, USA), and 5 µL of SureSelect all exon capture library, according to the Agilent SureSelect Target Enrichment protocol. Hybridization was performed at 65 °C for 24 h in a thermal cycler lid, with the temperature set to 105 °C. The captured DNA was washed and amplified. The final purified product was quantified by qPCR using the KAPA Library Quantification Kit (Roche, Basel, Switzerland) for Illumina Sequencing Platforms and qualified using the TapeStation D1000 ScreenTape (Agilent, Santa Clara, CA, USA). Sequencing was performed on the NovaSeq6000 platform (Illumina, San Diego, CA, USA).

### 4.3. Sequencing Data Processing

Integrated primary data analysis was conducted using Illumina’s Real-Time Analysis (RTA, v.3.4.4) software to generate raw images and base calling. The base calling files were converted into FASTQ format using bcl2fastq (v.2.20.0), with the demultiplexing option (barcode mismatches) set to 0. Quality control of raw sequencing was performed using FastQC. Paired-end sequences produced by the NovaSeq Instrument were mapped to the human reference genome using the BWA-MEM algorithm (BWA). The mapping results were generated in BAM format, excluding unordered sequences and alternate haplotypes. PCR duplicates were removed using MarkDuplicates.jar from the Picard-tools package, which identifies duplicate reads in starting positions after coordinate sorting. Base Quality Score Recalibration (BQSR) was applied to recalibrate the BAM files, using machine learning to empirically model sequencing errors and adjust the quality scores accordingly.

Variant genotyping for each sample was performed based on previously generated BAM files using the Haplotype Caller tool from GATK to identify SNPs and short indel candidates at nucleotide resolution. We filtered the variants using the VariantFiltration tool from GATK, which is designed for hard-filtering variant calls based on certain criteria. Records were filtered by modifying the FILTER field, and the filtered records were preserved in the output unless their removal was specified in the command-line parameters. The filtered variants were annotated using SnpEff and filtered using dbSNP and 1000 Genomes Project SNPs. The format of the final product was VCF. Subsequently, an in-house program and SnpEff were used to annotate additional databases, including ESP6500, ClinVar, dbNSFP, and ACMG.

### 4.4. Variant Calling Analysis and Protein–Protein Interaction Analysis

To identify different mutations in Korean and Caucasian cohorts, a variant calling analysis was performed on BCC marker genes from European and Asian cohorts. In addition to the Asian tissue samples, PBMC data for the corresponding tissue samples were collected. To focus exclusively on mutations in the skin tissue, the intersection of mutations in Asian PBMC and skin samples was excluded from the analysis. To compare mutation calls between Caucasian and Korean cohorts, we used BCFtools isec (v.1.20) [48]. Subsequently, to identify genes with significant mutations in the non-exposed group compared to the exposed group within the Korean cohort, we conducted an analysis using PLINK (v.1.90b6.21) [49]. Mutational profiles were annotated and visualized using MutationMapper in the cBioPortal (version 6.3.3) [50,51,52]. PPI analysis was conducted using STRING (version 12.0), revealing a network cluster between BCC marker genes and the two mutated genes in the non-exposed group [28].

## 5. Conclusions

Our study provides insights into the unique mutational landscape of BCC in the Korean population, revealing key differences from previously reported Caucasian cohorts. We identified distinct oncogenic mutations in critical Hh pathway genes, including *PTCH1* and *SMO*, as well as in tumor suppressor genes, such as *TP53* and *NOTCH2*. We also discovered population-specific differences in the mutational spectra of BCC.

Importantly, we found that *PTCH1* and *NOTCH1* may play central roles in BCC pathogenesis, regardless of UV exposure, as some mutations in these genes were observed only in the non-exposed groups. Our findings suggest that while UV radiation is a well-established risk factor for BCC, genetic alterations in specific pathways, particularly Hh and *NOTCH* signaling, may contribute to tumorigenesis in non-UV-exposed cases.

Additionally, we identified two novel genes, *TAS1R2* and *ADCY10*, as potential contributors to BCC pathogenesis, particularly in BCC of non-exposed areas. These genes have previously been linked to cancer; however, their specific roles in BCC development remain unclear. Their interaction with known BCC driver genes, such as *NOTCH2* and *TP53*, highlights the need for further functional studies to determine their biological relevance.

Our study highlights the population-specific genetic differences in BCC and suggests that *PTCH1* and *NOTCH1* may serve as key drivers of UV-independent tumorigenesis. Further functional studies and larger cohort analyses are needed to fully elucidate the molecular mechanisms underlying BCC in non-exposed areas and explore potential targeted therapeutic approaches based on these findings. Further analysis should establish ethnic-specific BCC diagnostic algorithms as distinct patterns observed in the Korean population. Comprehensive studies comparing molecular profiles between different ethnic groups are essential to develop medical implications and approaches tailored to Asian genetic backgrounds and environmental factors.

## Figures and Tables

**Figure 1 ijms-26-06941-f001:**
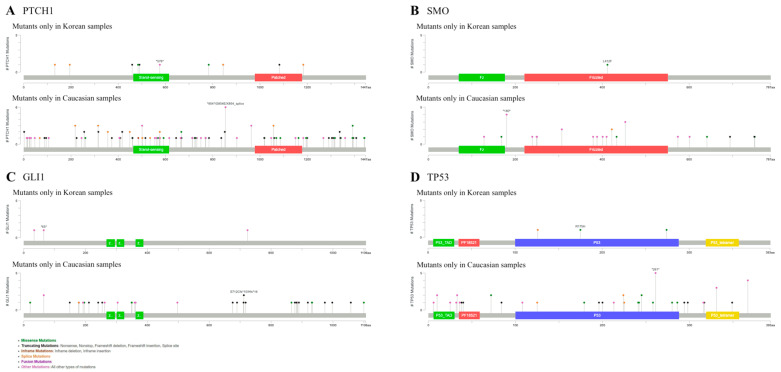
Different mutational profiles in Korean and Caucasian BCC groups. (**A**) Mutations of *PTCH1*. (**B**) Mutations of *SMO*. (**C**) Mutations of *GLI1*. (**D**) Mutations of *TP53*.

**Figure 2 ijms-26-06941-f002:**
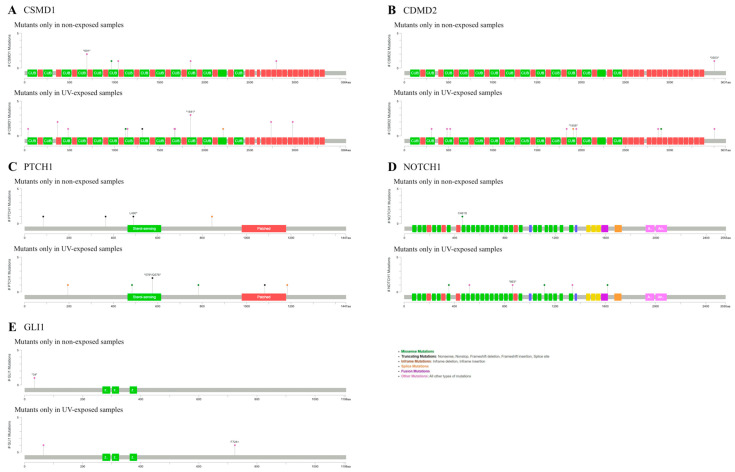
Different mutation profiles in the BCCs of exposed and non-exposed areas. (**A**) Mutations in *CSMD1*. (**B**) Mutations in *CSMD2*. (**C**) Mutations in *PTCH1*. (**D**) Mutations in *NOTCH1*. (**E**) Mutations in *GLI1*.

**Figure 3 ijms-26-06941-f003:**
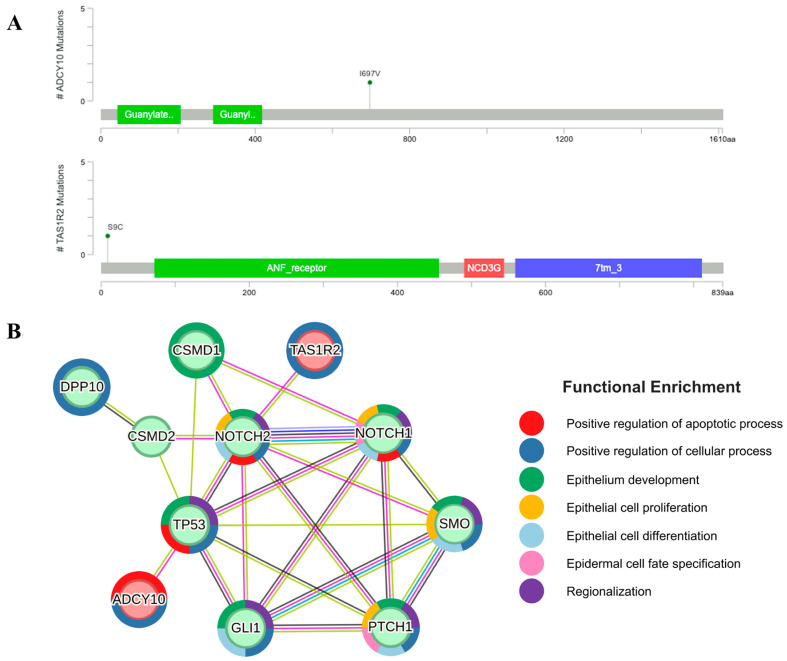
(**A**) Mutational profiles of *ADCY10* and *TAS1R2*. (**B**) Protein–protein interaction network between *ADCY10*, *TAS1R2* and BCC marker genes, and the functional enrichment of the network.

**Table 1 ijms-26-06941-t001:** Demographic features of the patients and clinicopathologic characteristics of the BCCs.

	Exposed (*n* = 8)	Non-Exposed (*n* = 6)	All (*n* = 14)
Age (mean ± SD)	65.38 ± 8.42	70.33 ± 7.26	67.50 ± 8.06
Sex, n (%)			
Male	6 (75.0)	3 (50.0)	9 (64.3)
Female	2 (25.0)	3 (50.0)	5 (35.7)
Ethnicity			
Korean	8 (100)	6 (100)	15 (100)
Skin type, n (%)			
III	6 (75.0)	5 (83.3)	11 (78.6)
IV	2 (25.0)	1 (16.7)	3 (21.4)
History of other NMSC, n (%)	0 (0.0)	0 (0.0)	0 (0.0)
History of sunburn, n (%)	0 (0.0)	1 (14.3)	1 (6.3)
Immunosuppression, n (%)	0 (0.0)	0 (0.0)	0 (0.0)
Anatomical site, n (%)			
Cheek	4 (50.0)	0 (0.0)	4 (28.6)
Nose	2 (25.0)	0 (0.0)	2 (14.3)
Eyelid	1 (12.5)	0 (0.0)	1 (7.1)
Neck	1 (12.5)	0 (0.0)	1 (7.1)
Back	0 (0.0)	2 (33.3)	2 (14.3)
Abdomen	0 (0.0)	2 (33.3)	2 (14.3)
Scalp	0 (0.0)	1 (16.7)	1 (7.1)
Calf	0 (0.0)	1 (16.7)	1 (7.1)
Histopathological subtype, n (%)			
Nodular	6 (75.0)	3 (50.0)	9 (64.3)
Superficial	2 (25.0)	3 (50.0)	5 (35.7)
Pigmentation, n (%)	8 (100)	6 (100)	14 (100)

SD, standard deviation.

## Data Availability

The data presented in this study are available upon request from the corresponding author. Public data used in this study were obtained from the NCBI BioProject repository with the accession PRJNA731355, https://www.ncbi.nlm.nih.gov/Traces/study/?acc=PRJNA731355.

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
