# Peer review of "Ethnic-Specific and UV-Independent Mutational Signatures of Basal Cell Carcinoma in Koreans"

_ijms, 2025, doi:10.3390/ijms26146941_

Round 1
Reviewer 1 Report
Comments and Suggestions for Authors
comments
-1) in abstract, you report "genes unique to non-UV-exposed tumors were further analyzed." report examples
2) line : linking them to TP53 and NOTCH2?? genes or proteins?
3) apart from molecular profile of the BCC in koreans, are there ant other korean -specific characteritisc of BCCs_ for example BCC in asians are more likely to present with pigmented dermoscopy structures comapred to Caucasians- report those traits in intro
4) nice images and figures
5) can those findings have clinical implications for exampel with medicactions that target Hedgehog pathways
6) systemic implications of sunexposure and BCC occurence ( 10.3390/cancers17040703)and thinking of further reseacrh based on race-specific criteria such as Koreans should also be reported
Author Response
1. In abstract, you report “genes unique to non-UV-exposed tumors were further analyzed.” Report examples.
Response1: We added “with protein-protein interaction analysis” for the report examples.
2. Line : linking them to TP53 and NOTCH2? Genes or proteins?
Response2: The links between TAS1R2, ADCY10, TP53, and NOTCH2 were analyzed on protein level via protein-protein interaction analysis.
3. apart from molecular profile of the BCC in Koreans, are there ant other Korean -specific characteristics of BCCs_ for example BCC in Asians are more likely to present with pigmented dermoscopy structures compared to Caucasians- report those traits in intro
5. can those findings have clinical implications for example with medications that target Hedgehog pathways
Response3 and 5: Regarding points 3 and 5, while these are important clinical considerations, our findings do not provide direct evidence to explain the prevalence of pigmented BCC in Asians or differential responses to Hedgehog inhibitors. Our study primarily focused on gene discovery, and the genes identified have not been previously reported in the literature as being directly linked to pigmentation or differential drug response in the context of BCC. Furthermore, this study did not include functional assays that would be necessary to elucidate such mechanisms.
6. systemic implications of sun-exposure and BCC occurrence (10.3390/cancers17040703) and thinking of further research sed on race-specific criteria such as Koreans should also be reported
Response6: In discussion, the above reference was cited as " While the impact of UV-exposure and BCC occurrence in regarding of systemic implications was presented by Karampinis et al., the mechanism beyond BCC in non-UV-exposed area is not well-known”. The further research direction was added in last part of conclusions, regarding race-specific criteria.
Reviewer 2 Report
Comments and Suggestions for Authors
This manuscript investigates the mutational signatures of basal cell carcinoma (BCC) in Koreans. The authors identified driver gene mutations specific to the Korean population. Moreover, they found that BCCs in non-ultraviolet (UV)-exposed tumor areas exhibited recurrent and novel mutations. The authors highlight unique ethnic-specific and UV-independent mutational profiles in Korean BCCs and suggest the need for population-specific precision oncology. The following points should be addressed:
- The resolution of all figures is low and too small to understand.
- The authors should label panels a–e in Figures 1 and 2.
- The mutational signature should be analyzed using SigProfiler and compared.
- I recommend that the authors add a table summarizing the mutational profiles and corresponding tumor sites.
Author Response
- The resolution of all figures is low and too small to understand.
- The authors should label panels a–e in Figures 1 and 2.
Response 1 and 2: We revised all figures, including main and supplementary figures, with higher resolution with label panels.
- The mutational signature should be analyzed using SigProfiler and compared.
Response3: We analyzed mutational signature using SigProfiler, finding that Caucasian samples presented much more signature in SBS for C>T alterations. C>T alterations are well-known consequences for UV exposure. This finding is consistent with our result and implies that Caucasian samples had more impacts from UV-exposure regarding to single base substitution signature. The figures of SigProfiler result are included in the attached word file.
- I recommend that the authors add a table summarizing the mutational profiles and corresponding tumor sites.
Response 4: We added the mutational profiles of UV-exposed site in revised supplementary table.

Round 2
Reviewer 1 Report
Comments and Suggestions for Authors
the authors did take my comments into account and the manuscript is improved
Reviewer 2 Report
Comments and Suggestions for Authors
The authors have addressed the points I previously noted.